# High-content fluorescence imaging with the metabolic flux assay reveals insights into mitochondrial properties and functions

Andrew Charles Little [1,2,7], Ilya Kovalenko[3,6,7], Laura Elaine Goo[1,2], Hanna Sungok Hong[2,3], Samuel Andrew Kerk [2,3], Joel Anthony Yates [1,2], Vinee Purohit [2,3], David Benner Lombard[2,4], Sofia Diana Merajver[1,2] & Costas Andreas Lyssiotis [2,3,5✉]

Metabolic flux technology with the Seahorse bioanalyzer has emerged as a standard technique in cellular metabolism studies, allowing for simultaneous kinetic measurements of respiration and glycolysis. Methods to extend the utility and versatility of the metabolic flux assay would undoubtedly have immediate and wide-reaching impacts. Herein, we describe a platform that couples the metabolic flux assay with high-content fluorescence imaging to simultaneously provide means for normalization of respiration data with cell number; analyze cell cycle distribution; and quantify mitochondrial content, fragmentation state, membrane potential, and mitochondrial reactive oxygen species. Integration of fluorescent dyes directly into the metabolic flux assay generates a more complete data set of mitochondrial features in a single assay. Moreover, application of this integrated strategy revealed insights into mitochondrial function following PGC1a and PRC1 inhibition in pancreatic cancer and demonstrated how the Rho-GTPases impact mitochondrial dynamics in breast cancer.

[1] Department of Internal Medicine, Division of Hematology and Oncology, University of Michigan, Ann Arbor, MI 48109, USA. [2] Rogel Cancer Center, University of Michigan, Ann Arbor, MI 48109, USA. [3] Department of Molecular and Integrative Physiology, University of Michigan, Ann Arbor, MI 48109, USA. [4] Department of Pathology and Institute of Gerontology, University of Michigan, Ann Arbor, MI 48109, USA. [5] Department of Internal Medicine, Division of Gastroenterology and Hepatology, University of Michigan, Ann Arbor, MI 48109, USA. [6]Present address: Insitro Inc, South San Francisco, CA 94080, USA. [7]These authors contributed equally: Andrew Charles Little, Ilya Kovalenko. ✉email: clyssiot@med.umich.edu

A primary output of cellular metabolism is chemical energy in the form of adenosine triphosphate (ATP). The major bioenergetic pathways that generate ATP in a cell are glycolysis and mitochondrial respiration[1,2]. Production of ATP from the mitochondria is coupled to the generation of reducing power in the tricarboxylic acid (TCA) cycle, and the respiration-dependent formation of a proton gradient by the electron transport chain (ETC).

While ATP generation is the most well-described role of the mitochondria, this multi-purpose organelle performs several additional important cellular functions[3–7]. Metabolic outputs of mitochondrial metabolism directly regulate signal transduction, gene expression, and cell death. Indeed, it is now established that mitochondrial metabolism impacts numerous physiological and pathophysiological states, including development, cancer, immune recognition and surveillance, and blood glucose control, to name a few[6,8–12]. The widespread influence of mitochondria in health and disease underscores the importance of continued development of strategies to fully characterize mitochondrial metabolism and function.

While recent emerging technologies have permitted more precise examination of mitochondrial functions and properties, each of these techniques are typically performed independently[13,14]. This is problematic in the sense that mitochondria are rapidly undergoing (bio)chemical, morphological, physiochemical, thermodynamic, and other changes at any given time; making experiment-to-experiment comparisons challenging. Therefore, capturing bioenergetic and functional data in a single multifunctional assay has the potential to yield greater, more controlled, and more precise mitochondrial information. Here, we describe an integrated platform that utilizes bioenergetic profiling technology alongside imaging of mitochondrial functions and properties to obtain a richer data set from a single experiment.

Metabolic flux technology using the Seahorse Bioanalyzer has emerged as an industry standard to assess the bioenergetic state of cells in vitro/ex vivo. It simultaneously measures pericellular pH and oxygen concentration in media as a function of time. These measurements provide a baseline surrogate for (1) glycolytic activity, which can be calculated from the extracellular acidification rate (ECAR), a readout of media pH reflecting lactic acid and bicarbonate accumulation, and (2) mitochondrial respiration or oxygen consumption rate (OCR), as determined by the extracellular oxygen level[15,16]. In addition, metabolic flux technology can also provide information on the bioenergetic properties and functional status of mitochondria. For example, mitochondrial poisons can be used to infer the bioenergetic flexibility of a cell, activity of ETC complexes, and maximal respiration capacity. Indeed, the simplicity, convenience, robustness, and sensitivity of the metabolic flux assay has made it a technology of choice for many laboratories[17–21].

Despite the widespread use of the Seahorse Bioanalyzer technology, acquisition of reliable data requires effective normalization strategies to correct for cell density. Multiple normalization methodologies have been used with varying degrees of acceptance by the research community. Examples include normalization to post-assay protein harvest or post-assay cell counting, normalization to pre-assay cell counting[22], or normalization via one of a variety of chemical colorimetric or fluorometric readouts (e.g., MTT, ATPGlo, WST-1). Specifically in a recent study employing small-interfering RNA (siRNA)/ short-hairpin RNA (shRNA) screening, Hoechst nuclei staining coupled with automated nuclei count was demonstrated to have better performance than other normalization methods[23]. Indeed, this strategy has been recently incorporated into the Seahorse pipeline to more adequately control for cell number with the

merger of BioTek Cytation5 and Seahorse XF assay platforms[24]. Herein, we optimize and extend this previous work, as nuclei staining can also be applied to determine cell cycle distribution[25,26]; an important cellular characteristic that affects bioenergetics. Importantly, mitochondrial bioenergetics have been previously shown to coordinate with cell cycle dynamics[27,28], further supporting the use of nuclei counter-staining in conjunction with the metabolic flux assay.

The use of nuclear stains can be valuable not only for cell counting, but also for more robust image focusing and calculation of regions of interests in a distance constrained fashion (e.g., 10 μm from nuclei mask); allowing quantification of secondary fluorescent signals at single-cell resolution. For instance, we have applied fluorescent staining of mitochondria via the MitoTracker Red cell dye and use proximity to the nucleus to quantify mito-chondrial content. MitoTracker Red is actively sequestered and retained in mitochondria[29], a process that is initially dependent on an intact mitochondrial membrane potential[30,31]. In addition to basic identification and quantification of mitochondria, we utilize MitoTracker dye in combination with high-content ima-ging to analyze mitochondrial fragmentation as an added feature integrated into our analysis pipeline[31,32].

Additional fluorescence-based dyes are similarly available to measure discrete mitochondrial parameters, including inner mitochondrial membrane potential ($\Delta\psi_m$) and mitochondrial reactive oxygen species (mtROS). $\Delta\psi_m$ is generated by the proton pumping complexes of the ETC. The energy "stored" in the $\Delta\psi_m$ is ultimately used to drive ATP production by complex V. While moderate fluctuations in the $\Delta\psi_m$ can reflect normal functioning of mitochondria, sustained increases or drops can lead to mito-chondrial pathology and/or target mitochondria for degradation. To build in the detection of $\Delta\psi_m$ into our imaging platform, we utilized the fluorescent dye tetramethylrhodamine ethyl ester (TMRE). TMRE is sequestered in the matrix of active mito-chondria based on the positive charge of the dye[33–35]. Depletion of the $\Delta\psi_m$ leads to loss of polarity and thus loss of dye seques-tration and signal.

Mitochondria are one of the primary sources of reactive oxygen species (ROS), which have been characterized to play important roles in physiological and pathophysiological processes[36–38]. The partial reduction of oxygen by the ETC leads to the formation of superoxide, a potent mitochondrial ROS (mtROS). MitoSOX is a mitochondrially targeted fluorescent dye[13] that is oxidized spe-cifically by superoxide to ethidium. The ethidium then inter-calates into mitochondrial DNA and thus produces fluorescence proportional to mtROS[39].

In this report, we describe a method that integrates the analysis of mitochondrial bioenergetics with mitochondrial properties, by implementing a variety of chemical fluorescent stains and high-content imaging into the Seahorse metabolic flux assay. This includes image analysis for cell number normalization and cell cycle distribution, along with mitochondria quantity, localization, fragmentation, membrane potential, and mtROS (Fig. 1). We established the utility of this novel methodology in human breast and pancreatic cancer cell lines using a variety of pharmacological probes, including those that perturb nuclear content and mito-chondrial functions, and respiration deficient pancreatic cancer cells. Then, we further extended the utility of our platform by interrogating mitochondrial content and function following genetic knockdown of the mitochondrial master regulatory pro-teins PGC1a and PRC1. Finally, application of this strategy yielded insights into the role of Rho-GTPases in mitochondrial dynamics in breast cancer. While the roles of the Rho-like family of Rho-GTPases in breast cancer progression were well char-acterized, to our knowledge, our study is the first to determine their role in the regulation of mitochondrial content,

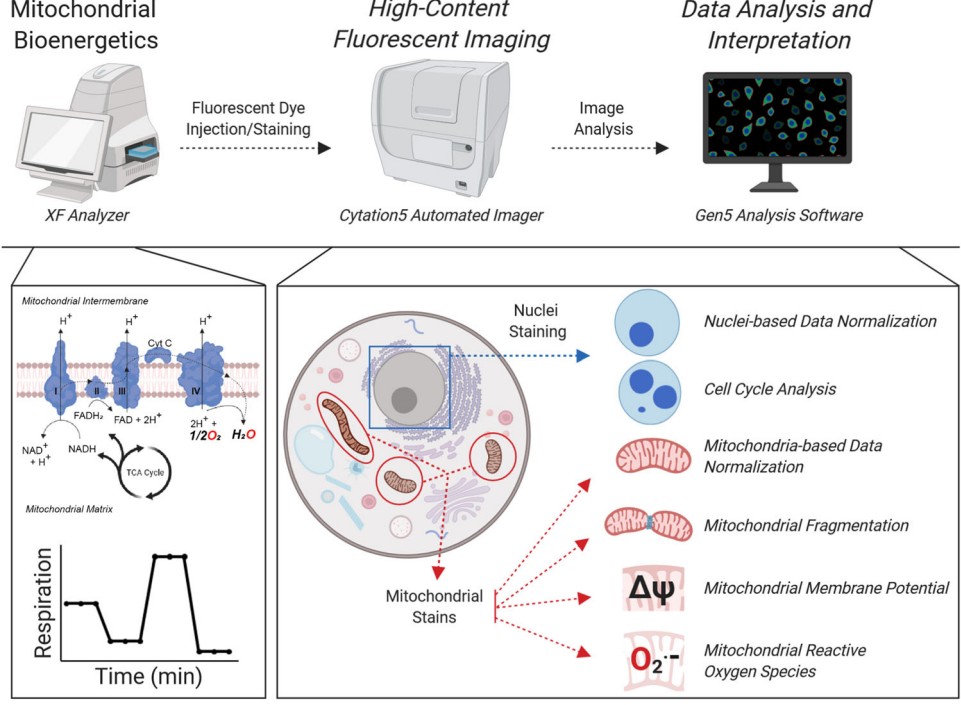

**Fig. 1 Platform to integrate the metabolic flux assay with high-content imaging.** Schematic overview of the integration of the metabolic flux assay with high-content fluorescent imaging and data analysis workflow. At the instrument level, cells are processed using the Seahorse metabolic flux assay and immediately stained with a variety of nuclear and mitochondrial dyes, which is completely integrated in the Seahorse bioanalyzer assay. The plates are then abstracted and imaged on a Cytation5 Automated Imager for downstream image analysis and interpretation. At the biochemical level, the metabolic flux assay provides OCR and ECAR data and information on other mitochondrial bioenergetic properties (by Mito Stress Test). The cells are then stained with nuclear and mitochondrial dyes that provide information on the cellular properties noted.

fragmentation, and respiratory capacity[40–43]. Collectively, this study enhances the utility of the metabolic flux assay and provides a more complete platform to study mitochondrial biology from multiple dimensions, simultaneously.

## Results

**Integration of nuclear imaging with the metabolic flux assay.**
*Normalization of cell number*: The Seahorse metabolic flux assay is a rapid and robust methodology to measure OCR and ECAR of living cells in culture. Owing to the sensitivity of the Seahorse XF analyzer to measure small changes in OCR and ECAR, it is critical that data are adjusted to account for well to well variability in cell number. To this end, we set forth to develop a high-content fluorescent imaging-based strategy using nuclear staining to quantify cell number directly after the metabolic flux assay (Fig. 1). In this iteration of our platform, we first run the Seahorse Mito Stress Test assay, in which OCR is measured at baseline and then following sequential administration of mitochondrial poisons from the instrument ports. After completion of the assay, we deliver the nuclear staining Hoechst dye via the fourth, and otherwise empty, port to live cells. The plate is then washed, and nuclei are counted on a Cytation5 Cell Imaging Multi-Mode Reader.

Initially, to compare the accuracy of our nuclei identified through image analysis versus cell seeding densities at plating, we seeded serial diluted T3M4 pancreatic cancer cells (from 1500 to 50,000 cells/well) in XF96-well plates for Seahorse analysis, nuclei counterstaining, and fluorescent imaging. As expected, we observe increases in raw OCR and ECAR values with increasing cells seeded (Fig. 2a Supplementary Fig. 2A, B). When normalizing OCR values according to cells seeded, we observe appreciable variation (Fig. 2b). Additionally, we aimed to determine if this strategy would be permittable for cells in

suspension cultures and implemented our pipeline on CD4 + T-cells. Nuclei or mitochondrial unit-based normalization strategies were not shown to impact OCR normalization versus standard cell counting strategies (Supplementary Fig. 2B). This is potentially due to ineffective nuclei and/or mitochondrial identification as fluorescent imaging of suspension cultures is inherently challenging to obtain single-cell resolution. Additional experimental approaches would be required to ascertain the utility of our approach in suspension-based cell systems. In contrast, cell normalization using fluorescently labeled nuclei offered more consistent OCR values in the 3000 to 50,000 cell-seeded range for adherent cells (Fig. 2c). We do not observe variation in normalized ECAR values between the cell counting and nuclei labeling strategies with increasing cell densities (Supplementary Fig. 2C, D).

Furthermore, when plotting OCR values per nuclei relative to the plate column, we noted discrepancies in data acquired from wells at the edge of the XF96-well plate (Fig. 2d). We hypothesized that this resulted from unequal distribution of cells within the well, and in particular, near to the center of the well where the Seahorse microchamber measures oxygen concentration (Supplementary Fig. 3A). Indeed, we found that cells in the wells at the perimeter of the plate are more likely to accumulate at the edge of the wells during the centrifugation process, thus artifactually lowering OCR values. As we were unable to correct for this artifact using nuclei counting, we employ the interior columns/rows of the Seahorse plate for experimentation. Similarly, for assay well normalization by nuclei counting, we utilize the cells in the center of the well (schematically represented in Supplementary Fig. 3B).

*Cell cycle analysis*: Nuclei staining intensity can be used to infer the stage of the cell cycle[25,26]. Therefore, we sought to determine

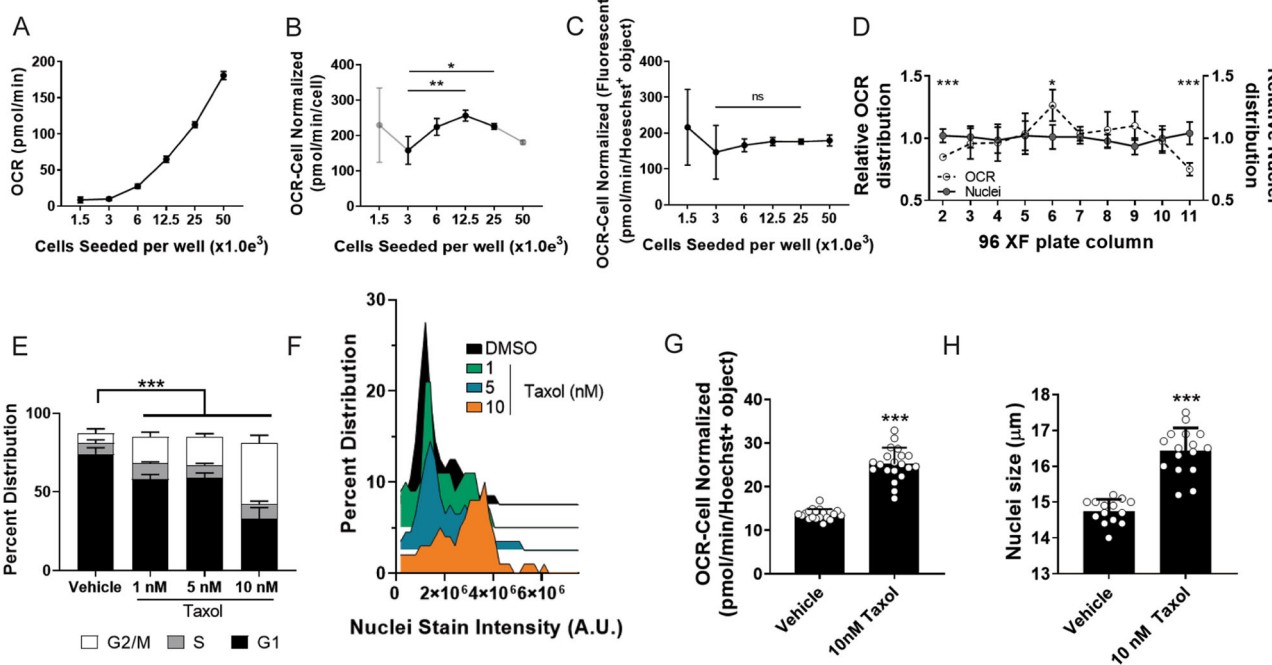

**Fig. 2 Integration of fluorescent-based nuclear imaging with the Seahorse metabolic flux assay. a** T3M4 cells were seeded in increasing densities in the Seahorse XF plate and processed through the XF assay for OCR and plotted. $n = 8$–10 biologically independent samples. **b** OCR values were corrected for and plotted against manually counted cell seeding densities. Statistical significance determined by one-way ANOVA; *$p = 0.038$, **$p = 0.001$. $n = 7$–10 biologically independent samples. **c** OCR measurements were corrected for fluorescently counted nuclei (i.e., Hoechst+ object) and plotted against seeded cell densities. $n = 8$ to 10 biologically independent samples. **d** OCR data (empty circle) extrapolated from individual columns of the XF plate display significant variation at the edge columns (i.e., columns 2 and 11), while no changes are observed in nuclei distribution (gray circle). Statistical significance determined by two-way ANOVA; **$p = 0.008$, ***$p = 0.000013$, ***$p = 5 \times 10^{-7}$. $n = 8$ biologically independent samples. **e** Cell cycle analysis distribution of taxol treated cells, identified through nuclei fluorescent imaging (e.g., Hoechst+ object). Statistical significance determined by one-way ANOVA, ***$p < 0.001$ (complete p-values can be seen in Supplementary Table 3). Data presented as the average +/− s.d. of $n = 10$ biologically independent samples. **f** Post hoc distribution analysis of cells treated with increasing concentrations of taxol and their respective nuclei staining intensity (A.U. arbitrary fluorescent units) $n = 10$ biologically independent samples. **g** Image analysis of nuclei size via fluorescently labeled nuclei post taxol treatment; $n = 20$ biologically independent samples. **h** OCR values corrected for fluorescently identified nuclei are increased post taxol treatment; $n = 14$ to 16 biologically independent samples. Statistical significance in **g**, **h** determined by Student's t-test; **g** ***$p = 10^{-14}$; **h** ***$p = 4 \times 10^{-9}$. All experiments were the result of ≥2 independent experiments. All measurements were obtained from distinct samples.

if we could integrate this analysis to the XF-Cytation5 platform. To this end, we treated T3M4 pancreatic cancer cells across a dose range of taxol, an FDA approved chemotherapeutic (e.g., Paclitaxel), which arrests cells in G2/M phase of the cell cycle due to its microtubule polymerization inhibiting properties[44,45]. Of note, at the doses examined taxol exhibited cell cycle arrest without toxicity (Supplemental Fig. 4A, B). Nuclear staining intensity was then collected and used to demonstrate that taxol induces a dose-dependent accumulation of cells in the G2/M phase of the cell cycle (Fig. 2e, f). Similarly, at the highest dose examined, we also observed an increase in nuclear size (Fig. 2h), demonstrating another utility for our imaging platform. Finally, we observed increased OCR in T3M4 cells that have been treated with taxol, a previously described feature of taxol administration[25] (Fig. 2g). These data support the utility of nuclei counterstaining and imaging to assess cell cycle and nuclear size post metabolic flux assay.

**Integration of mitochondria imaging with metabolic flux assay.** OCR data provide a readout for total respiration of the cells within a well. This is impacted by both the number and activity of the mitochondria in the cells. The latter is related in part to the structural properties of the mitochondria, where fused mitochondrial networks tend to be more respiratory than fissed mitochondria[46,47]. The location of mitochondria within a cell has

also been reported to impact their function[48,49]. Furthermore, heterogeneity in mitochondrial density, structure, location, and function also exists on a cell to cell basis within a population of cells in a well. Therefore, we hypothesized that a more complete understanding of this information would provide considerable utility in accurately characterizing mitochondrial function and OCR output from the metabolic flux assay. We therefore adapted our high-throughput imaging platform to capture these parameters by incorporating a series of mitochondrial dyes, delivered in a manner akin to the delivery of the nuclear dye described above, followed by image analysis.

*Normalization of OCR data to mitochondrial content*: We set forth to image mitochondrial content and localization and to evaluate the relationship between these parameters and OCR. To this end, we empirically tested a panel of Mitotracker dyes with T3M4 pancreatic cancer cells plated in the Seahorse 96XF cell plate. Mitotracker Deep Red was identified as the most consistent and robust probe to visualize mitochondria, as it readily retained bright fluorescence in live cell formats, as well as post fixation (Supplementary Fig. 5). Specifically, it exhibited the highest signal/background ratio in the Seahorse XF cell plate, was fixable, and readily integrates into our Seahorse workflow.

To demonstrate the utility of this approach, we generated a Rho0 (ρ0) S2-013 pancreatic cancer cell line (S2-013-ρ0) with severely depleted mitochondrial DNA resulting from prolonged

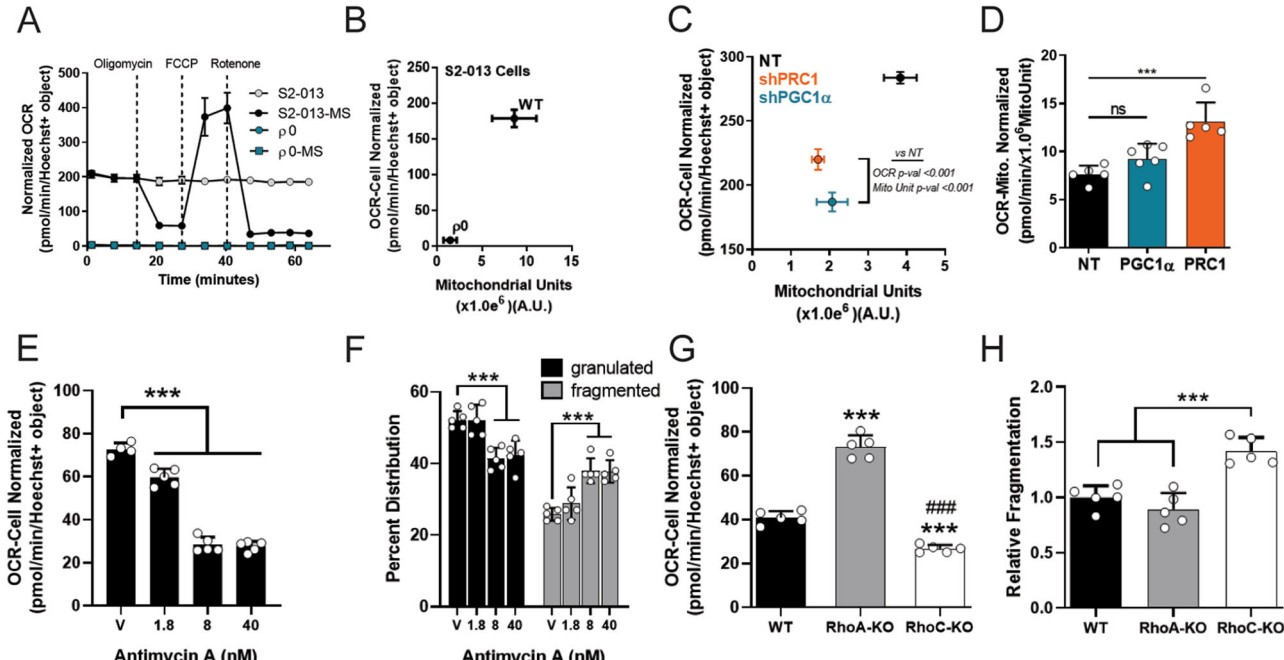

**Fig. 3 Analysis of mitochondrial quantity and fragmentation downstream of the metabolic flux assay. a** Hoechst+ nuclei corrected OCR data post mitochondrial stress test ("-MS" designation; e.g., S2-013-MS) or measurements of basal OCR values (i.e., S2-013) in wild type S2-013 cells or S2-013-ρ0 (Rho0) cells. $n = 6$ biologically independent samples for controls and $n = 20$ biologically independent samples subjected to mitostress test (i.e., "-MS"). **b** OCR values corrected for fluorescent nuclei plotted against total fluorescently identified mitochondria (i.e., MitoTracker positive arbitrary fluorescent units; A.U.). $n = 15$ to 18 biologically independent samples. **c** OCR values normalized for nuclei in shNT (non-targeting), shPRC1, or shPGC1α T3M4 cells, plotted against fluorescently identified mitochondria (MitoTracker positive A.U.) $n = 5$ biologically independent samples. NT vs. shPGC1α ***$p = 10^{-10}$; NT vs. shPRC1 ***$p = 1.3 \times 10^{-8}$. **d** OCR values in NT, shPRC1, or shPGC1α T3M4 cells corrected for total fluorescently identified mitochondria (e.g., Mito Normalized; MitoUnit). $n = 5$ biologically independent samples. Statistical significance determined by one-way ANOVA; n.s. non-significant; ***$p = 0.0002$. **e** Hoechst+ nuclei corrected OCR values post Antimycin A treatment. $n = 5$ biologically independent samples. Statistical significance determined by one-way ANOVA; veh vs. 1.8 nM ***$p = 10^{-6}$; for veh. vs. 8, 40 nM ***$p < 10^{-15}$. **f** Overall levels of granulated (black) or fragmented (gray) mitochondria post Antimycin A treatment. $n = 5$ biologically independent samples. Statistical significance determined by two-way ANOVA; ***$p < 0.001$. Complete statistical data can be seen in Supplemental Table 4. **g** Hoechst+ nuclei corrected OCR values of WT, RhoA KO, or RhoC KO VARI068 breast cancer cells. $n = 5$ biologically independent samples. Statistical significance determined by one-way ANOVA; WT vs. RhoC-KO ***$p = 2 \times 10^{-8}$; WT vs. RhoA-KO ***$p = 0.00013$; RhoA-KO vs. RhoC-KO ###$p = 4 \times 10^{-10}$. **h** Levels of fragmented mitochondria in WT or RhoA/C-KO VARI068 cells. $n = 5$ biologically independent samples. Statistical significance determined by one-way ANOVA; WT vs. RhoC-KO ***$p = 0.0006$; RhoA-KO vs. RhoC-KO ***$p = 0.00007$. All experiments were the result of ≥2 independent experiments. All measurements were obtained from distinct samples.

culture in ethidium bromide[50,51]. Depletion of mitochondrial DNA results in the loss of a functional ETC and thus the ability of mitochondria to maintain their $\Delta\psi_m$ or respire. As expected, S2-013-ρ0 cells completely lack a respiratory profile (Fig. 3a) and have greatly diminished Mitotracker Deep-Red staining (Fig. 3b and Supplementary Fig. 6A), the latter of which requires $\Delta\psi_m$ to be retained in the mitochondria. No changes in ECAR were detected in S2-013 WT vs. ρ0 cells (Supplementary Fig. 6B). These results, using an artificial system of mitochondrial depletion, illustrate the utility our imaging platform to capture information on mitochondrial content, as well as function.

To extend our imaging platform to a system with a less extreme mitochondrial defect, we employed a genetic approach to deplete mitochondria in a confined timeframe. Protein regulator of cytokinesis 1 (PRC1) and peroxisome proliferator-activated receptor gamma coactivator 1-alpha (PGC-1α) are master regulators of mitochondrial biogenesis[52–55]. We generated PRC1 and PGC-1α knockdown T3M4 cells using shRNA (see qPCR results of gene knockdown in Supplementary Fig. 7A, B). Loss of PRC1 or PGC-1α resulted in fewer overall mitochondria, as determined by Mitotracker fluorescent staining and quantification (Supplementary Fig. 7C). We then assayed PRC1 and PGC-1α knockdown cells for basal levels of OCR and normalized the data to nuclei number, followed the XF assay with Mitotracker

staining and quantification. Using the nuclei normalization strategy, we observed lower overall OCR values in PRC1 and PGC-1α knockdown cells as compared to their WT counterparts (Fig. 3c). In addition, we find fewer overall mitochondria in the PRC1 and PGC-1α knockdown cells, as expected (Fig. 3c).

In contrast, if we normalize our OCR data to mitochondrial content, which we define as MitoUnit, the normalized data reflects no changes in overall mitochondrial function in PGC-1α knockdown cells and increased OCR in PRC1 knockdown cells (Fig. 3d). Mitochondrial normalization is achieved by utilizing Mitotracker-based fluorescence imaging, rather than fluorescently imaged nuclei. This is useful when implementing studies to examine mitochondrial functionality in systems where mitochondrial counts may be impacted. In this example, our PRC1 and PGC-1α knockdown cells harbor functional, albeit fewer, mitochondria. Results such as these have the capacity to be easily misinterpreted if using standard normalization techniques. These data strongly support the utilization of mitochondrial quantification as a parallel normalization technique for OCR data in the metabolic flux assay, as the two normalization parameters provide different and important outputs.

*Mitochondrial fragmentation analysis*: Next, we sought to explore the use of Mitotracker staining as a method to evaluate mitochondrial fragmentation patterns. Mitochondrial fragmentation

has been observed in many settings, including apoptosis, responses to oxidative stress, neurodegeneration, among various others, and is an important feature of mitochondrial biology[56–59]. As a positive control, we treated MIA PaCa-2 pancreatic cancer cells with the ETC complex III inhibitor Antimycin A to induce fragmentation, a previously described feature of its administration[60]. As expected, we observed a decrease in OCR levels in a dose-dependent fashion post Antimycin A treatment (Fig. 3e). Using Mitotracker staining/imaging and the BioTek Gen5 spot analysis tool (see BioTek application notes for further detail), we were able to quantify decreases in granulated mitochondria and increases in fragmented mitochondria following Antimycin A treatment (Fig. 3f). These results confirmed the utility of mitochondrial image analysis using Mitotracker fluorescent dye for this purpose.

The Rho-like family of Rho-GTPases have been well characterized for their participation in the progression of breast cancer[40–43] and the alteration of metabolic phenotype[22]. Therefore, we next applied our imaging and analysis strategy to explore the potential for fragmented mitochondria in breast cancer cells that lack either the Rho-like family of Rho-GTPases RhoA or RhoC via CRISPR/Cas9 knockout (WT, RhoA KO, RhoC KO) (Supplementary Fig. 8A). First, we examined basal OCR values in the breast cancer patient-derived cell model VARI068[61] and followed with Mitotracker staining directly into the XF assay. Our data revealed elevated basal OCR levels in the RhoA KO cell line and lower OCR levels in the RhoC KO cell line (Fig. 3g). No changes in mitochondrial fragmentation/elongation were observed in the more respiratory RhoA KO cells; in contrast, greater levels of fragmented mitochondria were observed in the RhoC KO cells (Fig. 3h). We confirmed that changes in fragmentation were not due to altered overall levels of mitochondria (Supplementary Fig. 8B). These data illustrate that quantification of mitochondrial fragmentation via Mitotracker imaging is a robust and useful method that can be readily integrated into the metabolic flux assay to enrich the overall data set regarding mitochondrial features.

*Fluorescence staining for mitochondrial membrane potential*: To build detection of $\Delta\psi_m$ into our imaging platform, we utilized the fluorescent dye TMRE. First, we demonstrated that TMRE dye fluorescence correlated with increased OCR values, indicative of active mitochondria (Fig. 4a). Uncouplers of $\Delta\psi_m$, such as FCCP, dissipate the proton gradient. At low concentrations of FCCP, the ETC competes to maintain a proton gradient and induce maximal respiration. At higher concentrations of FCCP, dissipation of the proton gradient outpaces the capacity of the ETC to maintain a gradient, poisoning the mitochondria, and thereby impairing respiration and $\Delta\psi_m$. Treatment of T3M4 cells with FCCP in a dose-dependent fashion demonstrated this expected rise and fall in OCR (Fig. 4b and Supplementary Fig. 9A). Similarly, loss of OCR correlated with a loss of TMRE uptake by mitochondria, as reflected by decreased TMRE intensity quantified from fluorescent imagine analysis (Fig. 4c); suggesting impaired mitochondrial function.

*Mitochondrial reactive oxygen species fluorescence imaging*: Under physiological conditions, the biogenesis and quenching of ROS regulate various cellular processes and are tightly controlled[62]. Excessive ROS production, on the other hand, is implicated in numerous pathologies, including aging and cancer. Superoxide is the ROS produced from the mitochondria by incomplete oxidation of oxygen during respiration. Therefore, we sought to include detection of mitochondrial superoxide alongside the metabolic flux assay. To measure mitochondrial superoxide, we used the mitochondrially targeted MitoSOX dye. To induce mitochondrial superoxide production, we treated the pancreatic cancer cell line PA-TU-8902 with the membrane-

targeted radical initiator ditertbutyl peroxide (DTBP) or hydrogen peroxide ($H_2O_2$). We then monitored OCR levels, integrated MitoSOX dyes into the metabolic flux assay, and observed mitochondrial superoxide via MitoSOX fluorescence. We observed that both $H_2O_2$ and DTBP treatment had a dose-dependent inhibitory effect on OCR levels (Fig. 4d). These data are consistent with prior literature demonstrating that oxidative stress can lead to diminished basal OCR values[63,64]. Following DTBP or $H_2O_2$ stimulation, expectedly we quantified a marked increase in MitoSOX fluorescence (Fig. 4e) (representative images can be seen in Supplemental Fig. 9B). Data corrected for overall changes in OCR versus increases in MitoSOX fluorescence can be observed in Fig. 4f. Collectively, these data confirm the utility of MitoSOX dye incorporation into the XF assay, providing robust quantifiable data regarding mitochondrial respiration and the generation of ROS.

## Discussion

Here, we present a robust strategy to integrate high-content, high-throughput fluorescent imaging into the Seahorse metabolic flux assay. This integrated approach aims to build upon prior efforts to utilize fluorescence-based counterstains for normalization of bioenergetic data[65], rather than relying on rudimentary cell number correction strategies. Our pipeline enables the evaluation of select features in one complete experiment on a single XF96 well plate, increasing the utility and output of a single experiment while minimizing the potential for plate-to-plate variability. We initiated this strategy to address observed well to well inconsistencies in the OCR bioenergetic parameter, as similarly reported by others[24,66]. While classic approaches to normalize mitochondrial bioenergetic data with total cellular input or quantified cellular protein may be a quick and efficient, we found these to be inconsistent across experiments and cell lines. Furthermore, as we detail in Fig. 2, the location of cells in a well of a Seahorse plate impacts OCR. By accounting for both the number of cells (based on nuclear staining), and more specifically, the number of nuclei in proximity to the sensors, our normalization method reduces variability across columns and wells. Furthermore, we employed nuclear size and staining intensity to provide information into cell cycle dynamics.

Extension of OCR normalization to total mitochondria provides further resolution and more granular information about mitochondrial capacity, as illustrated in Fig. 3. Staining for total mitochondria is easily achieved through the inclusion of Mitotracker dyes or biosensors into the metabolic flux assay workflow. We consider this an important experimental control, as a wide array of experimental conditions may impact mitochondrial biogenesis or mitochondrial content, which we have demonstrated affect OCR results. Correction of metabolic flux data using this method provides additional information alongside nuclei-based cell counting. However, the use of Mitotracker dyes in this pipeline are not limited to their use as a normalization tool for metabolic flux data. We also found that the Mitotracker staining patterns readily allow for the identification and quantification of fragmented mitochondria (Fig. 3f, h). Indeed, Mitotracker staining could be further amended to quantitatively describe subcellular mitochondrial localization, organelle or molecular co-localization, fission-fusion dynamics, mitochondrial shape (e.g., sphericity), among others.

We then applied this platform in two targeted studies. First, we demonstrated that, while knockdown of two master regulators of mitochondrial biogenesis PGC1a and PRC1[52–55] did result in less cellular respiration, this was the result of fewer mitochondria, not less active mitochondria (Fig. 3c, d). This illustrates the utility of building in a mitochondrial normalization strategy into the

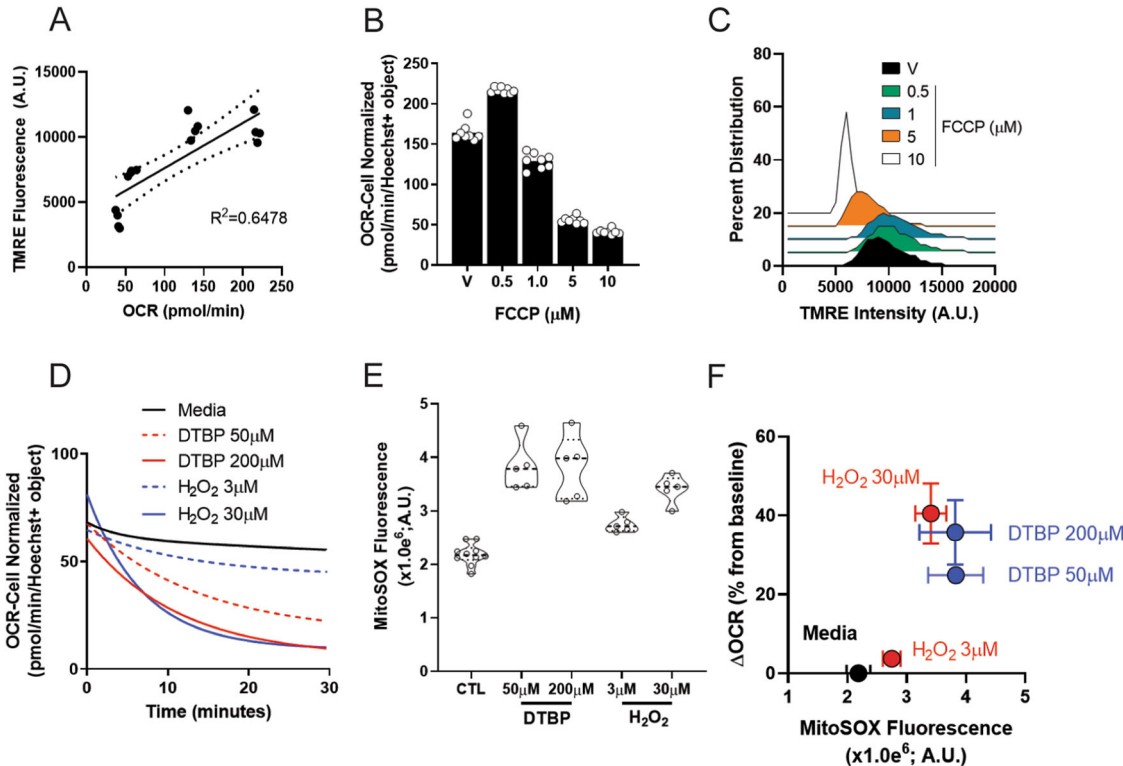

**Fig. 4 Analysis and quantification of $\Delta\psi_m$ and mitochondrial ROS. a** TMRE fluorescence plotted with respect to OCR in T3M4 cells. **b** OCR readout following treatment across a dose range of FCCP in T3M4 cells. $n = 8$ biologically independent samples. **c** Distribution analysis of TMRE fluorescence intensity (A.U.) in T3M4 cells. $n = 50$ biologically independent samples per bin. **d** Exponential curve fit of OCR data of PA-TU-8902 pancreatic cancer cells treated with either DTBP (ditertbutyl peroxide; red lines) or hydrogen peroxide ($H_2O_2$; blue lines). $n = 5$ biologically independent samples. **e** Violin plots displaying induction of MitoSOX fluorescence (A.U.) following either DTBP or $H_2O_2$ treatment. $n = 5$ to $10$ biologically independent samples. **f** Multi-analysis plot displaying change in OCR values plotted against MitoSOX fluorescence in PA-TU-8902 cells post DTBP or $H_2O_2$ treatment. $n = 5$ biologically independent samples. All experiments were the result of $\geq 2$ independent experiments. All measurements were obtained from distinct samples.

metabolic flux assay workflow. We then examined how the Rho-like family of Rho-GTPases impact mitochondrial fragmentation, based on their known role in regulating cytoskeletal dynamics, and, more specifically, our previous work, which found that inhibition of RhoC impairs the metabolic properties of inflammatory breast cancer cells[22]. Application of the MitoTracker strategy downstream of the metabolic flux assay revealed that RhoC deletion increases the number of fragmented mitochondria in inflammatory breast cancer cells, which we hypothesize contributes to the depleted OCR levels (Fig. 3g, h). Future studies are required to fully characterize the Rho-like Rho-GTPases and their regulation of mitochondrial dynamics.

The decision to utilize TMRE and MitoSOX dyes were based on the idea of synthesizing a broad data set of mitochondrial features that can be analyzed downstream of mitochondrial respiration on the same cells in the same well. A limitation of the TMRE stain is the requirement for live cell imaging. Subsequent to this, the cells can be stained with MitoTracker, MitoSOX and Hoechst, fixed, and entered into the downstream normalization and analysis workflow. Indeed, we selected a series of dyes with non-overlapping wavelengths (Supplementary Fig. 1) so that this suite of parameters can be simultaneously analyzed on a limited number of wells. As noted, MitoTracker, MitoSOX and nuclear staining with Hoechst are amenable to fixation and retain their subcellular localization, and we have had success storing plates for analysis months later. Accordingly, Seahorse plates can be imaged immediately after analysis (post dye addition) or stored for downstream analysis if large numbers of plates are being prepared for high-throughput screening. It is also important to note that

cells imaged for MitoTracker, TMRE, and MitoSOX cannot be the same cells on which the Mito Stress Test is performed. Requisite for the Mito Stress Test analysis is the use mitochondrial poisons that impact respiration, membrane potential, oxidative stress, among others. Thus, the staining and analysis protocols need to be performed in parallel wells.

We also anticipate that the use of dyes to monitor cellular events beyond those described herein could be readily appended to this workflow. For example, dyes are available to analyze lysosomal labeling (e.g., LysoTracker), mitochondrial calcium signaling (e.g., Fura-2, Fluo-3), and/or other oxidative stress parameters (e.g., CellROX)[67]. Similarly, genetically encoded sensors exist to monitor changes in pH, ATP, redox cofactors (e.g., NADH, NADPH), and oxidative stress[68–72], and these could be applied to this workflow. Indeed, the Cytation5 imaging system permits the use of various filter cube sets allowing visualization of many different fluorescent channels. Finally, this platform also has the potential to be used for drug screening, as approaches to disrupt mitochondrial metabolism for cancer therapy, to facilitate mitochondrial function in aging and other mitochondrial disease, or to promote the turnover of damaged mitochondria in degenerative conditions remain focal points in the development on novel therapeutics[73–76].

In addition to those noted above, there are other limitations associated with the experimental platform described. Our approach utilizes an array of fluorometric reporter dyes. If not properly controlled, such dyes harbor the potential to provide artifactual data. This is especially noted for the use of TMRE, where high concentrations of the dye can yield false positives; i.e.,

TMRE directly impacts mitochondrial membrane potential. Potential artifacts can be observed with Mitotracker (i.e., based on cellular sulfhydryl redox status) or MitoSOX (i.e., cellular pH fluctuations). We provide examples, and encourage the inclusion, of positive and negative controls for each of these dyes. Similarly, it is recommended that dye concentration be empirically determined on an experimental platform basis.

Furthermore, the use of alternate methods to support data acquired from fluorescent reporters should also be considered when designing these types of experiments for verification of results. As an example, a western blot analysis of total mitochondrial content provides a suitable verification of the Mitotracker mitochondrial content analysis. The use of high-resolution microscopy (e.g., confocal microscopy) is an appropriate orthogonal approach to better characterize mitochondrial morphology to support findings from our pipeline described herein. In other words, our platform was designed to provide the user a high-throughput pipeline to examine various mitochondrial features in a single experiment. It should not replace more thorough evaluations of mitochondrial characteristics, and we strongly suggest that users perform additional experimentation to confirm findings gleaned using our assay.

In total, our platform provides high-resolution normalization strategies for Seahorse data that encompass nuclei and mitochondrial based fluorescent imaging and quantification. We define the inclusion of additional mitochondrial stains to generate a robust data set, in one high-throughput experiment, characterizing mitochondrial biology in a continuous kinetic fashion. This is imperative in studying mitochondria, as they can rapidly change phenotypes in response to their environment. Thus, capturing the greatest amount of data in a continuous, rapid manner will provide more consistent results and may reveal additional mitochondrial characteristics otherwise not captured in single experiment formats.

## Methods

**Cell culture**. MIA PaCa-2 were obtained from ATCC. Pa-Tu-8902 cells were obtained from DSMZ. S2-103 and T3M4 cells were generously provided by Anguraj Sadanandam (ICR, UK). VARI068 cells were generated from a patient-derived xenograft in the Merajver laboratory. PA-TU-8902, MIA PaCa2, S2-013, and T3M4 pancreatic cancer cell lines were cultured in Dulbecco's Modified Eagle's Medium (DMEM) supplemented with 10% FBS. S2-013 $\rho^0$ cells were supplied additionally with 10 µg/ml EtBr, 100 µg/ml pyruvate and 50 µg/ml uridine. The VARI068 breast cancer cell line was maintained in RPMI1640 supplied with 10% FBS. Cell lines were STR profiled and routinely tested for mycoplasma.

**Chemicals and probes**. FCCP, oligomycin, rotenone, antimycin A, TBH70X, tert-Butyl hydroperoxide solution (Luperox), taxol, and Poly-L-Lysine (mol wt 70,000–150,000, 0.01%) were obtained from Sigma. Hoechst 33342, Mitotracker DeepRed, MitoSOX, and SYTOX Green were from ThermoFisher Scientific. TMRE was from Abcam. All compounds were stored at −20 °C except Taxol and Hoechst (4 °C). Dyes were stored protected from light. FCCP, Rotenone, Antimycin A, Hoechst, MitoTracker DeepRed, and SYTOX were stored at −20 °C, TMRE was pre-diluted at 10 µM in media (10×) and aliquoted for single use. MitoSOX was likewise aliquoted for single use[30].

**shRNA constructs and viral transduction**. pLKO lentiviral vectors were ordered as bacterial glycerol stocks from Sigma, MISSION® shRNA Bacterial Glycerol Stock, cat# SHCLNG-NM_013261. The shNT sequence was subcloned into pLKO backbone vector. The references for the sequences are provided in Supplementary Table 1. Viral particles were produced by the University of Michigan Vector Core. T3M4 cells were transduced with the addition of polybrene (Sigma) to 8 µM final concentration. Cells were selected with 1 µg/ml puromycin (Sigma) for 3 days. Following selection, transduced T3M4 cells were seeded at 20,000 cells/well for the metabolic flux-imaging assay as described below. Remaining cells were processed for qPCR.

**RNA isolation and qPCR**. $10^6$ cells were lysed and RNA isolated using the RNAeasy kit (Qiagen) according to the manufacturer's instructions. One microgram of total RNA was added for cDNA synthesis using the iScript cDNA Synthesis Kit (Bio-Rad) following the manufacturer's protocols. For qPCR, Fast SYBR Green Master Mix (ThermoFisher Scientific) was used, and amplification was detected with an Applied Biosystems QuantStudio3 Real-Time PCR System. The sequences of the primers used for the amplification are provided in the Supplementary Table 2.

**Metabolic flux assays**. Adherent cells were seeded at $2 \times 10^4$ cells/well in normal growth media (cell line specific) in a Seahorse XF96 Cell Culture Microplate. To achieve an even distribution of cells within wells, plates were rocked at 25 °C for 20–40 min. For each staining group, one extra well on the outer perimeter of the plate was seeded to calibrate image acquisition parameters. The plate was then incubated at 37 °C overnight to allow the cells to adhere. The following day, growth media was exchanged with Seahorse Phenol Red-free DMEM and either basal OCR was measured (for wells that were to be imaged with mitochondrial dyes, see Imaging section below) or an XF Cell Mito stress test (Agilent) was performed according to the manufacturer's instructions. In both cases, the last injection port was used for cell stain/dye injection. Upon completion of the Seahorse assay, cells were washed three times with pre-warmed phenol-free DMEM media (no FBS) and transferred to the Cytation5 for imaging.

For cells grown in suspension, $10^5$ cells/well were added to a poly-lysine coated Seahorse XF96 Cell Culture Microplate in Phenol Red-free DMEM-based assay media. The plate then was centrifuged at $300 \times g$ for 30 min with gentle acceleration and deceleration. The plate is then rotated 180° and the centrifugation repeated. Immediately following completion of the centrifugation, OCR was measured or an XF Cell Mito stress test was performed followed by imaging in the Cytation5 as above; schematic of assay workflow can be visualized in Supplemental Fig. 1A.

**High-content imaging**. Imaging was carried out using a Cytation5 Cell Imaging Multi-Mode Reader (BioTek, VT, USA). The environment was controlled at 5% $CO_2$ and 37 °C. Hoechst (1 µg/ml final concentration) was imaged using a 365 nm LED in combination with an EX 377/50 EM 447/60 filter cube. SyTOX Green was imaged using a 465 nm LED in combination with an EX 469/35 EM 525/39 filter cube. TMRE (1 µM final concentration) and MitoSOX Red (5 µM final concentration) were imaged using a 523 nm LED in combination with an EX 531/40 EM 593/40 filter cube. MitoTracker DeepRed (200 nM final concentration) was imaged using a 623 nm LED in combination with an EX 628/40 EM 685/40 filter cube. Dyes were delivered at the end of the XF Cell Mito stress test from Port D at 10x to the entire plate. Cytation5 image excitation/emissions spectra utilized for imaging of various fluorescent stains are depicted in Supplemental Fig. 1B. Image analysis was completed using Gen5 software (BioTek). Example of Gen5 nuclei masking and secondary fluorescent signal masking algorithms can be observed in Supplemental Fig. 1C.

**Statistics and reproducibility**. For each independent experiment, methods of statistical analysis are defined in the representative figure legend. For each analysis, a minimum of two biological replicates (independent experiments; new passage of cells) were performed, with each harboring ≥3 technical replicates; please see figure legend for specifics. For all analyses, dependent on multivariable correction or lack thereof, one/two-way analysis of variance (ANOVA) or standard Student's t-test was applied to evaluate statistical significance.

**Reporting summary**. Further information on research design is available in the Nature Research Reporting Summary linked to this article.

## Data availability
Primary data is readily available upon request from corresponding author, C.A.L. Data storage is redundantly archived on local and network-based applications. Primary data is accessible via the cloud-based University of Michigan storage systems at any time per reasonable request. Data for figures is present as Supplementary Data for figures.

## Code availability
No specialized or custom code was utilized in capturing data for this study. Algorithms used to analyze imaging data are publicly available and built into the BioTek Gen5 image analysis software.

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

## Acknowledgements

We would like to thank Dr. John Dishinger of BioTek for his assistance with designing imaging and image analysis protocols. We wish to thank Prof. Youle RJ for providing the YFP-Parkin plasmid and Dr. Stefan Prechtl and Dr. Sven Christian for initial proof-of-concept experiments. Fig. 1 and Supplementary Figure 1 were created using BioRender.com. C.A.L. was supported by a Pancreatic Cancer Action Network/AACR Pathway to Leadership award (13-70-25-LYSS); Junior Scholar Award from The V Foundation for Cancer Research (V2016-009); Kimmel Scholar Award from the Sidney Kimmel Foundation for Cancer Research (SKF-16-005); a 2017 AACR NextGen Grant for Transformative Cancer Research (17-20-01-LYSS); an ACS Research Scholar Grant (RSG-18-186-01); NIH-R37-CA237421 and NIH-R01-CA248160. D.L. was supported by R01GM101171. A.C.L. and S.D.M. were supported by the Breast Cancer Research Foundation. C.A.L. and S.D.M. were supported by the Rogel Cancer Center core grant NIH-P30-CA046592-29.

## Author contributions

Conceptualization: A.C.L., I.K., C.A.L.; methodology: A.C.L., I.K., J.A.Y.; investigation: A.C.L., I.K., L.E.G., H.S.H., S.A.K., J.A.Y., V.P.; visualization: A.C.L., I.K.; formal analysis: A.C.L., I.K.; manuscript and figure preparation: A.C.L., I.K., J.A.Y., C.A.L.; resources: D.L., S.D.M., C.A.L.; supervision: S.D.M., C.A.L.; funding acquisition: S.D.M., C.A.L.

## Competing interests

At the time of manuscript preparation I.K. owned stock in Agilent Technologies (manufacturer of Seahorse and Cytation instruments) and was employed by Sartorius North America, the manufacturer of IncuCyte imaging instruments. I.K. and C.A.L. are holders of a patent describing the method in this manuscript. I.K. and C.A.L. declare no involvement of their financial interests in design or interpretation of the results of this study. All additional authors declare no financial and non-financial interests.
