## [Peer Review file · Communications Biology]

Reviewers' comments:

Reviewer #1 (Remarks to the Author):

In this study, Little and colleagues couples the metabolic flux assay from the Seahorse bioanalyzer (OCR and ECAR) to fluorescent based imaging. Using the Cytation 5 automated imager for high throughput image analysis and allowed for a more stable and consistent normalization of the oxygen consumption rates to cell density. Of particular interest, is the incorporation of mitochondrial fluorescence stains into the seahorse metabolic flux assay workflow. The authors demonstrate that observing mitochondrial features in tandem with OCR measurements provides a more comprehensive data set, allowing for further insight into mitochondrial contribution to oxygen consumption rates.

Overall, I find the work to be well executed with positive controls included in the experiments elucidating mitochondrial function (fragmentation, and potential). Additionally, the limitation of study section makes clear for having proper controls as well when using the mitochondrial dyes and alternative methods to support the characterization of the mitochondrial function.

There are some minor points I was wondering if the authors might address:

1. A text change and reference to the introduction (line 61) where ECAR measurement was equated to glycolysis may not be entirely correct. Mookerjee et al showed TCA cycle activity producing CO₂ can be converted to HCO₃⁻ which can contribute to ECAR measurements significantly (PMID:28270511). A more thorough introduction on the Seahorse bioanalyzer measurements and recent conversations of ECAR and OCR to ATP production rates would help.
2. A more careful read-through of the paper to make sure the figures referenced in the text is correct. I've picked up a few. Line 251 reference of Figure 2G and Line 254 reference of Figure 2H. Additionally, line 428 reference of figure 3K and 3L doesn't exist.
3. I was wondering if the authors have data for the reliability of mitochondrial fluorescence imaging in other cell types. For example in T cells or dendritic cells where the cell size are smaller and may be more difficult to pick up fragmented mitochondria with the Cytation 5 software and imager. I noticed this was done in CD4+ T cells (Figure S2B) but I feel a more in depth discussion and representative images would really strengthen the paper.

Eric Ma, PhD

Van Andel Institute

333 Bostwick Ave. N.E., Grand Rapids, Michigan 49503-2518

Phone: 616 - 234 - 5838

Reviewer #2 (Remarks to the Author):

This is a fine technical study that provides new insight into a platform that could have wide-reaching impact on the study of metabolism. I have no comments to enhance it.

Reviewer #3 (Remarks to the Author):

Brief summary of the manuscript

The author's describe the merger of BioTek Cytation5 and Seahorse XF assay platforms to study the coordinated biology of mitochondrial bioenergetics and cell cycle dynamics. The data presented intends to show evidence of increased consistency in results for Seahorse XF analyzer data, while revealing additional mitochondrial characteristics using a variety of normalization techniques.

Overall impression of the work

The paper presents background research in a clear and concise order. The experiments performed show data that compliment the research question; does coupling metabolic flux assays with high-content fluorescence imaging provide a means for normalizing mitochondrial respiration data using various normalizing factors? And, the conclusions presented in the paper remain within the limits of the findings.

Specific comments, with recommendations for addressing each comment

Please consider the following: (1) add cited studies from Martin Brand and David Nichols to provide relevant background information, specifically in the field of mitochondrial bioenergetics; (2) please also consider adding citations for previous key findings related to using tetramethylrhodamine ethyl ester (TMRE) to study mitochondrial membrane potential (line 94 – 103).

Although I appreciate, the degree of the work involved to generate this data, one main concern that the authors can likely clear up with further explanation includes supplemental figure 2C and 2D. Supplemental figure 2C and 2D appear different when compared visually side by side; although the author states that “We do not observe significant variation in normalized ECAR values between the cell counting and nuclei labeling strategies with increasing cell densities (lines 230 and 231). Please consider taking an alternative statistical analytic approach to compare ECAR values normalized by cell counting and ECAR values normalized by nuclei labeling.

Please address the significance of the additional experiments that include the knockdown of PGC1a and PRC1. The use of the term “MitoUnit” needs further explanation and clarification. In line 319 the authors say that these results “could be easily misinterpreted using traditional normalization strategies.” Please include references for these ‘traditional normalization strategies.’

The authors might consider adding a brief account of the importance of PGC1a and PRC1 to mitochondrial biogenesis that they reference in the discussion (lines 418 – 419). I was challenged to find literature that supports PRC1 role as a “master regulator” of mitochondrial biogenesis; please address.

Also please include references for the “Potential artifacts [that] can be observed with Mitotracker, MitoSox, and TMRE in the section of the manuscript labeled “Limitations of the Study” beginning at line 466.

Overall I agree that capturing bioenergetics and functional data in a single functional assay can potentially provide greater insight into mitochondrial biology. I look forward to your comments and thank you for the chance to review this manuscript.

We are grateful for the reviewers' time, thoughtful feedback, careful attention to detail, and this functional peer review process. By addressing the comments, we have made for a stronger and clearer manuscript. Below we provide a point-by-point rebuttal to the referee comments (referees' comments in plain text and our responses in red). In the accompanying revised manuscript text, our updates are similarly presented in red.

REVIEWERS' COMMENTS:

Reviewer #1: In this study, Little and colleagues couple the metabolic flux assay from the Seahorse bioanalyzer (OCR and ECAR) to fluorescent based imaging. Using the Cytation 5 automated imager for high throughput image analysis and allowed for a more stable and consistent normalization of the oxygen consumption rates to cell density. Of particular interest, is the incorporation of mitochondrial fluorescence stains into the seahorse metabolic flux assay workflow. The authors demonstrate that observing mitochondrial features in tandem with OCR measurements provides a more comprehensive data set, allowing for further insight into mitochondrial contribution to oxygen consumption rates.

Overall, I find the work to be well executed with positive controls included in the experiments elucidating mitochondrial function (fragmentation, and potential). Additionally, the limitation of study section makes clear for having proper controls as well when using the mitochondrial dyes and alternative methods to support the characterization of the mitochondrial function.

Eric Ma, PhD
Van Andel Institute
333 Bostwick Ave. N.E., Grand Rapids, Michigan 49503-2518
Phone: 616 - 234 - 5838

Response: We appreciate the supportive comments and thoughtful suggestions from Dr. Ma, which we have addressed in full below.

1. "A text change and reference to the introduction (line 61) where ECAR measurement was equated to glycolysis may not be entirely correct. Mookerjee et al showed TCA cycle activity producing CO₂ can be converted to HCO₃⁻ which can contribute to ECAR measurements significantly (PMID:28270511). A more thorough introduction on the Seahorse bioanalyzer measurements and recent conversations of ECAR and OCR to ATP production rates would help."

We appreciate Dr. Ma's suggestion, and we agree entirely that our description of ECAR was incomplete. Please see lines 61-65 in the text, where we update the manuscript text to accurately describe the major metabolite contributions to ECAR and cite the above noted reference.

2. "A more careful read-through of the paper to make sure the figures referenced in the text is correct. I've picked up a few. Line 251 reference of Figure 2G and Line 254 reference of Figure 2H. Additionally, line 428 reference of figure 3K and 3L doesn't exist."

We thank Dr. Ma for his careful attention to detail. We have addressed the above noted typos, and carefully re-read through the manuscript in its entirety.

3. "I was wondering if the authors have data for the reliability of mitochondrial fluorescence imaging in other cell types. For example in T cells or dendritic cells where the cell size are smaller and may be more difficult to pick up fragmented mitochondria with the Cytation 5 software and imager. I noticed this was done in CD4+ T cells (Figure S2B) but I feel a more in depth discussion and representative images would really strengthen the paper."

We thank the Dr. Ma for this suggestion. We have made changes to include raw cell count data to compare against nuclei or mitochondrial based normalization strategies in supplemental figure 2B. Additionally, we updated the text to account for normalization in suspension cells (like T cells), see lines 233-241. Collectively, we did not find that our approach was useful in suspension cultures. Moreover, this may be in part to the lack of resolution obtainable from widefield fluorescence microscopy under these conditions.

Reviewer #2: This is a fine technical study that provides new insight into a platform that could have wide-reaching impact on the study of metabolism. I have no comments to enhance it.

Response: We appreciate the supportive comments from the reviewer and are pleased to hear that s/he considers it a valuable contribution to the literature.

Reviewer #3: Brief summary of the manuscript. The author's describe the merger of BioTek Cytation5 and Seahorse XF assay platforms to study the coordinated biology of mitochondrial bioenergetics and cell cycle dynamics. The data presented intends to show evidence of increased consistency in results for Seahorse XF analyzer data, while revealing additional mitochondrial characteristics using a variety of normalization techniques.

Overall impression of the work. The paper presents background research in a clear and concise order. The experiments performed show data that compliment the research question; does coupling metabolic flux assays with high-content fluorescence imaging provide a means for normalizing mitochondrial respiration data using various normalizing factors? And, the conclusions presented in the paper remain within the limits of the findings.

Response: We thank the reviewer for their time and instructive comments.

1. "add cited studies from Martin Brand and David Nichols to provide relevant background information, specifically in the field of mitochondrial bioenergetics"

We have added additional citations for Brand and Nicholls, who have had wide reaching contributions to the field of bioenergetics. (references 7, 12, 15, 16, 34)

2. "please also consider adding citations for previous key findings related to using tetramethylrhodamine ethyl ester (TMRE) to study mitochondrial membrane potential (line 94 – 103)"

We have added pertinent references related to TMRE-based testing of membrane potential. (references 33-35)

3. "one main concern that the authors can likely clear up with further explanation includes supplemental figure 2C and 2D. Supplemental figure 2C and 2D appear different when

compared visually side by side; although the author states that “We do not observe significant variation in normalized ECAR values between the cell counting and nuclei labeling strategies with increasing cell densities (lines 230 and 231). Please consider taking an alternative statistical analytic approach to compare ECAR values normalized by cell counting and ECAR values normalized by nuclei labeling.”

We apologize if this was unclear in the text. In order to determine if using additional data correction or normalization techniques would be required, we must confirm there is significant variability in the primary method, e.g. standard cell counting prior to assay plating. In this case, we did not see variable effects in ECAR when using standard cell counting methodology to normalize the data. Since this data is comparing several variables i.e. multiple seeding densities, we must correct for multiple comparisons. Therefore, we implemented ordinary one-way ANOVA. In this instance, there was not substantial change in ECAR values at the varying cell densities to pass significance. Below are the statistical testing parameters to display these results. Only one item passed significance testing in the nuclei labeled data set, comparing 6e3 vs 50e3 cell densities. Since our data defined that 50e3 cells seeded yielded too confluent of a cell monolayer for reliable normalization techniques, we are not including this concentration in our analyses.

Cell Count Normalized

Cells seeded per well x 1e3	Mean Diff.	95.00% CI of diff.	Significant?	Summary	Adjusted P Value
3 vs. 6	10.51	-8.894 to 29.92	No	ns	0.6019
3 vs. 12.5	12.15	-7.256 to 31.56	No	ns	0.4435
3 vs. 25	14.79	-4.614 to 34.20	No	ns	0.2318
3 vs. 50	8.926	-10.48 to 28.33	No	ns	0.7508
6 vs. 12.5	1.639	-17.77 to 21.05	No	ns	0.9999
6 vs. 25	4.280	-15.13 to 23.69	No	ns	0.9863
6 vs. 50	-1.586	-20.99 to 17.82	No	ns	0.9999
12.5 vs. 25	2.641	-16.77 to 22.05	No	ns	0.9986
12.5 vs. 50	-3.225	-22.63 to 16.18	No	ns	0.9963

Nuclei Count Normalized

Cells seeded per well x 1e3	Mean Diff.	95.00% CI of diff.	Significant?	Summary	Adjusted P Value
3 vs. 6	4.776	-7.243 to 16.80	No	ns	0.8467
3 vs. 12.5	9.989	-2.030 to 22.01	No	ns	0.1558
3 vs. 25	7.225	-4.795 to 19.24	No	ns	0.4889
3 vs. 50	-9.902	-21.92 to 2.118	No	ns	0.1628
6 vs. 12.5	5.213	-6.806 to 17.23	No	ns	0.7932
6 vs. 25	2.449	-9.571 to 14.47	No	ns	0.9904
6 vs. 50	-14.68	-26.70 to -2.658	Yes	**	0.0084
12.5 vs. 25	-2.765	-14.78 to 9.255	No	ns	0.9834

4. “Please address the significance of the additional experiments that include the knockdown of PGC1a and PRC1. The use of the term “MitoUnit” needs further explanation and clarification. In line 319 the authors say that these results “could be easily misinterpreted using traditional normalization strategies.” Please include references for these ‘traditional normalization strategies.’”

We thank the reviewer for this suggestion. We have now added text (lines 330-338) to bolster our description of mitochondrial units. Additionally, the corresponding references to

support our use of PGC1a and PRC1 as regulators of mitochondrial biogenesis are now cited.

5. “The authors might consider adding a brief account of the importance of PGC1a and PRC1 to mitochondrial biogenesis that they reference in the discussion (lines 418 – 419). I was challenged to find literature that supports PRC1 role as a “master regulator” of mitochondrial biogenesis; please address.”

Citations are now in place to support our statements regarding PGC1a and PRC1 as master regulators of mitochondrial biogenesis.